# Social and Demographic Factors Associated with Postnatal Depression Symptoms among HIV-Positive Women in Primary Healthcare Facilities, South Africa

**DOI:** 10.3390/healthcare9010065

**Published:** 2021-01-12

**Authors:** Kebogile Elizabeth Mokwena, Nontokozo Lilian Mbatha

**Affiliations:** Department of Public Health, School of Health Care Sciences, Sefako Makgatho Health Sciences University, Ga-Rankuwa 0204, South Africa; lilian.mbatha@gmail.com

**Keywords:** postnatal depression, HIV-positive women, social factors, demographic factors

## Abstract

Background: Mothers living with HIV are at risk for mental health problems, which may have a negative impact on the management of their HIV condition and care of their children. Although South Africa has a high prevalence of HIV, there is a dearth of studies on sociodemographic predictors of postnatal depression (PND) among HIV-positive women in South Africa, even in KwaZulu Natal, a province with the highest prevalence of HIV in the country. Objective: The objective of the study was to determine sociodemographic factors associated with the prevalence of postnatal depression symptoms among a sample of HIV-positive women attending health services from primary healthcare facilities in Umhlathuze District, KwaZulu Natal. Methods: A quantitative cross-sectional survey was used to collect data from 386 HIV-positive women who had infants aged between 1 and 12 weeks. The Edinburgh Postnatal Depression Scale (EPNDS), to which sociodemographic questions were added, was used to collect data. Results: The prevalence of PND symptoms among this sample of 386 HIV-positive women was 42.5%. The age of the mothers ranged from 16 to 42 years, with a mean of 29 years. The majority of the mothers were single or never married (85.5%; *n* = 330), living in a rural setting (81.9%; *n* = 316%), with a household income of less than R 2000 (estimated 125 USD) per month (64.9%; *n* = 120). The government child support grant was the main source of income for most of the mothers (53%; *n* = 183). PND symptoms were significantly associated with the participant’s partner having other sexual partners (*p*-value < 0.001), adverse life events (*p*-value = 0.001), low monthly income (*p*-value = 0.015), and being financially dependent on others (*p*-value = 0.023). Conclusion: The prevalence of PND symptoms among the sample is high, with a number of social and demographic factors found to be significantly associated with PND. This requires the consideration of sociodemographic information in the overall management of both HIV and postnatal depression. Addressing the impact of these factors can positively influence the health outcomes of both the mother and the baby.

## 1. Introduction

Global health literature continues to highlight the role and impact of mental health on overall health [1]. Although the literature reports the decline in mental health in the general population, the high prevalence of mental illness among people living with HIV is of great concern, as its occurrence is reported to be more frequent when compared with the HIV-negative population [2]. Sub-Saharan countries have the highest HIV prevalence, with 61% of global HIV new infections in 2018, with South Africa registering an estimated 4.7 million women aged 15 years and above that are HIV positive [3]. The reduction of HIV transmission through effective protocols for preventing mother-to-child transmission becomes imperative, thus the need to focus on maternal mental health among HIV-positive women.

Maternal mental health problems have emerged as a significant public health challenge. However, the focus on maternal health has been on reducing maternal and infant mortality with less focus on the promotion of maternal mental health [4]. The prevalence of postnatal depression (PND) worldwide ranges between 0.5% and 82.1% in developing and developed countries [5]. It is therefore predictable that PND in the perinatal spectrum is found to be a significant challenge which may impede parenting skills and proper functioning of a new mother [6,7]. The effects of PND concern not only the affected mother but also her children [8]. PND thus presents a major threat to the child’s mental health and psychological development [9]. Women with PND experience more negative emotions towards their infants and are less sensitively attuned to them, impairing mother–infant attachment and bonding [10].

Low socioeconomic status, which includes low education level and inadequate income due to unemployment, is a risk factor for PND [11,12,13]. While low education level is a significant risk factor for PND, especially in developing countries [14,15] being employed and a high level of education are protective factors [16,17,18,19]. However, some studies found no significant relationship between level of education or employment status and PND among HIV-positive women, [20], which suggests that the relationship between some socioeconomic factors and PND is complex and is confounded by other factors playing out in the mothers’ situation at a particular time.

There is a lack of consensus about the relationship between maternal age and PND, and while some studies report both young and older mothers at risk for PND [21,22], others have reported that women who were 25 years old and younger were found to be five times more likely to develop PND than older women [23].

Social support for women of childbearing age, which has been recognized as a stress resistance resource that contributes to positive adaptation, has been identified to be a protective factor for PND [22,24]. Family, friends, and others may become a source of emotional, informational, and physical support that women need during the perinatal period. Lack of social support is an important determinant of PND, particularly among HIV-positive women, and may adversely affect management and treatment outcomes of their HIV-related illnesses [3,6]. Given the benefit of partner support for maternal and infant health, the prevention of mother-to-child transmission program in South Africa strongly recommends partner involvement during the pregnancy of an HIV-positive woman [24], as the support of a partner is beneficial to the overall well-being of a pregnant woman who is also HIV positive [25].

Although a number of studies have reported that being married has a protective effect for PND, such benefits can only hold in a healthy relationship, as opposed to being married in a domestic violence situation [26,27]. Moreover, lower depression risks have been reported among women who were never married compared with those who were married (8), while a polygamous marriage is a risk factor associated with PND [28]. This implies that the quality of the relationship with the partner is crucial as a predictor of depression among women, as greater relational power and the availability of social support has a greater protective effect against PND than being married (8), and that being a single mother is better than being in an unsupportive relationship [29,30].

Within Sub-Saharan Africa, high intimate partner violence during the perinatal period is a major contributor to PND [26,31]. HIV-positive women in South Africa often experience a variety of intimate partner violence, which also impacts negatively not only their mental health but also their ability to adhere to Antiretroviral Therapy (ART). Pregnancy alone can be stressful, but it can be more stressful when there are stressful adverse life events during pregnancy such as the loss of a job, financial strain, or the death of a loved one, which can all contribute to the development of PND. The relationship between negative adverse life events and PND has been established [22,28,32].

Although there are several categories of risk factors for PND, this paper is limited to social and demographic factors, as the impact of these are more pronounced among HIV-positive women. The focus of this this paper is to understand the relationship of these factors in the context of HIV infection.

## 2. Research Methods and Design

### 2.1. Study Design

A cross-sectional survey was used to identify social and demographic predictors of PND among HIV-positive women who were attending primary healthcare facilities for postnatal care in the Umhlathuze Municipality.

### 2.2. Study Setting

Umhlathuze Municipality in the KwaZulu Province is the largest municipality in the King Cetshwayo district. The municipality has an estimated population of 885,944 and 13 primary healthcare facilities. The utilization of the health facilities is about 3.2 visits per person per year, with a postnatal clinic visit rate of 56% in 6 days (DHIS 2018/19).

### 2.3. Study Population and Sampling Strategy

The study population consisted of mothers who had infants aged 1–12 weeks, who were attending postnatal clinic care in the participating clinics. Studies have reported that postnatal depressive symptoms mostly manifest 2–3 months (8–12 weeks) post-natally among first-time mothers [33]. Data were collected in 10 of the 13 primary healthcare facilities. A survey of mothers who were attending postnatal services in the participating clinics on the dates of data collection, and who accepted the invitation to participate in the study, was conducted. Given an estimated 32,853 women who give birth annually in the district and using the Raosoft sample size calculator, a confidence level of 95%, 5% error margin, and a response rate of 50%, a minimum sample size of 380 was calculated.

### 2.4. Data Collection

The Edinburgh Postnatal Depression Scale (EPNDS) and a questionnaire was used to collect data. The EPNDS is a validated screening tool used to screen for PND symptoms and has been reported to be both reliable and valid for global use [34,35,36], including other African countries [37] and South Africa [28,38,39,40]. It consists of 10 self-reporting questions, which identify depression symptoms. Each question presents 4 answer choices scored on an ordinal scale of 0–3. The responses are summed to a possible total score of 0–30, with a greater score indicating increased severity of PND. The EPNDS score threshold varies, with some researchers using a cut-off of 10 [37]. However, higher cut-off scores are preferred as they yield greater specificity and sensitivity [40,41]. A higher threshold of 13 increases the sensitivity and specificity to 90.0% and 92.1%, respectively, with a positive predictive value of 98.9%, while a threshold of 11 yields acceptable specificity of 77%, a sensitivity of 80%, and a positive predictive value of 52.6% [39]. For this study, a cut-off of 12 was used, which is a middle ground that is likely to yield a specificity between 77% and 92.1%. Other studies in South Africa also used this cut-off value. The IsiZulu questionnaire was pretested on 15 women at a local clinic. The pretesting highlighted the need to rephrase a few questions to enhance the consistency of the responses. Data were collected by the first author and a research assistant over a period of four months (1 May to 30 August 2018). The language used was either English or IsiZulu and was determined by the preference of the individual participants.

The questionnaire consisted of sociodemographic questions and a section that measured PND using the Edinburgh Postnatal Depression Scale (EPNDS). Demographic questions included maternal age, race, marital status, place, religion, level of education, employment, monthly income, and source of income. Social information gathered included partner and family social support and history of adverse life events in the previous six months. HIV-related variables included the year of HIV diagnosis, diagnosis during pregnancy, ART intake during pregnancy, and disclosure of HIV status to family members and partner. The choice of variables was informed by the literature from previous PND studies conducted in South Africa [26,28,29].

### 2.5. Ethical Considerations

Ethical clearance was obtained from the Sefako Makgatho Health Sciences University Research Ethics Committee and from the Province of KwaZulu Natal Health Research and Knowledge Management Committee. Permission to conduct the study was obtained from the management of the health facilities, and written informed consent was obtained from individual participants.

### 2.6. Data Analysis

The data collected were captured on a Microsoft Excel 2016 spreadsheet, cleaned and verified, and then imported to STATA version 13 for analysis. Descriptive statistics included summary and frequency count for all sociodemographic-, HIV-, and social-related information. The Chi-square test was used to compute *p*-values. Logistic regression was used to determine sociodemographic- and HIV-related variables that are associated with PND symptoms, with statistical significance set at ≤0.05.

To determine social and demographic factors associated with PND symptoms, a staged approach to logistic regression analysis was used. Firstly, only demographic variables were included in the model. Secondly, social support variables that were found to be significantly associated with PND symptoms at bivariate analysis were included in the model (partner support, other support, currently in a sexual relationship, partner with other sexual partners, partner threats, and adverse life events). Lastly, statistically significant HIV-related variables were added.

## 3. Results

### 3.1. Description of the Sample

A total of 386 HIV-positive mothers attending Umhlathuze primary healthcare facilities for postnatal service, who had a baby aged between 1 and 12 weeks, participated in the study. The age of the mothers ranged from 16 to 42 years with a mean age of 29 (standard deviation = 5.8) years. Two hundred and nineteen (56.7%) were within the age range of 26–35 years. The majority of the mothers were single or never married (85.5%; *n* = 330), living in a rural setting (81.9%; *n* = 316%), Christians (72%; *n* = 278), completed matric (71.5%; *n* = 276), unemployed (78.8%; *n* = 304), and earning less than R 2000 (estimated 125 USD) per month (66%; *n* = 122). The legal minimum wage for a fulltime worker in South Africa in 2020 is R 3520 (220 USD) monthly, which suggests that the earnings of such participants fall below the minimum wage or they do not have fulltime employment. The government child support grant of R 740 or 46 USD per child was the main source of income for most mothers (47.4%; *n* = 183). The government child support grant is offered to lower-income households to assist parents with the costs of the basic needs of their child, and this indicates that most mothers are from lower-income households. Income is an important indicator of socioeconomic status or standard of living, and more than half (52.1%) did not indicate the household monthly income in the self-completed questionnaires. The rest of the demographic and social descriptions of the participants are shown in Table 1 and Table 2 below.

Most of the participants were in a relationship (87.6%; *n* = 338), with 28.2% (*n* = 109) reporting that their partners had other sexual partners. The majority (66.6%, *n* = 257) were living with their parents, and the majority (88.7%; *n* = 338) were receiving financial support from their partners. Although the majority were never threatened by their partners (89.4%, *n* = 328), 10.6% (*n* = 39) reported being threatened with physical violence by their partners. Experiences of adverse life events over the previous six months were reported by 36.3% (*n* = 139) of the participants.

### 3.2. HIV-Related Information

Thirty-five percent of the participants (*n* = 135) were diagnosed with HIV within the previous 12 months, with 58.5% (*n* = 223) reporting that HIV was diagnosed during the last pregnancy. About a third did not know their partner’s HIV status, and a fifth (19.8%; *n* = 76) did not disclose their HIV-positive status to any family member. Table 3 below shows the rest of the HIV-related information about the participants.

### 3.3. Prevalence of PND

The EPNDS scores ranged from 0 to 30, with a mean of 10.7 (SD = 6.1). A cut-off of ≥12 was used as a threshold for determining the likelihood of PND, and 42.5% (*n* = 164) had EPNDS scores of 12 or greater, thus meeting the threshold for postnatal depression (see Figure 1 below).

PND symptoms were reported by a greater proportion of participants who had completed matric (63%; *n* = 104), earned less than R 2000 per month (45.8%; *n* = 22), had a partner and family as a source of financial support (53%; *n* = 78), knew the partner’s HIV status (53.3%; *n* = 82), knew their babies’ HIV status (47.5%; *n* = 103), were in a relationship (83%; *n* = 136), had partner financial support (83.9%; *n* = 135), had other forms of support (86%; *n* = 141), were never threatened by a partner (81.5%, *n* = 123), had partners who had other sexual relationships (45%; *n* = 67), and had experienced adverse life events in the past six months (56.5%; *n* = 91).

Univariate analysis showed that PND symptoms were statistically significantly associated with a number of sociodemographic variables, as reflected in Table 4 below.

### 3.4. Bivariate Analysis of Sociodemographic and Social Factors Associated with PND

From bivariate logistic regression analysis, PND symptoms among HIV-positive women were found to be significantly statistically associated with adverse life events (OR 4.7, 95% CI: 2.94–7.55, *p*-value = 0.00), threats from partner (OR 4.2, 95% CI: 1.96–9.76, *p*-value = 0.00), other sexual partners (OR 3.5, 95% CI: 2.19–5.59, *p*-value = 0.00), partner support (OR 2.9, 95% CI: 1.14–4.69, *p*-value = 0.01), monthly income (OR 2.8, 95% CI: 0.91–11.68, *p*-value = 0.05), other support (OR 2.3, 95% CI: 1.08–4.80, *p*-value = 0.01), currently in a relationship (OR 2.1, 95% CI: 1.08–4.06, *p*-value = 0.02), and income source (OR 1.5, 95% CI: 0.98–2.41). From bivariate logistic regression, variables which had a weak association with PND symptoms were marital status (OR 1.6), place of residence (OR 1.4), partner HIV status (OR 1.2), disclosure of HIV-positive status to family members (OR 1.1), ART difficulties (OR 1.4), and who the mother lives with (OR 1.7). Table 4 above shows the results of the bivariate analysis.

### 3.5. Logistic Regression Analysis of the Factors Associated with PND

A multiple logistic regression analysis was conducted to identify independent variables that are statistically significantly associated with PND (Table 5). The results showed that mothers whose partners have other sexual partners were almost five times more likely to report PND symptoms compared with those whose partners do not have other sexual partners (OR = 4.8, 95% CI: 1.55–14.97, *p*-value = 0.00). Mothers who have experienced at least one adverse life event in the past six months were found to be four times more likely to report PND symptoms than those who did not report any adverse events in the past six months (OR = 4.4, 95% CI: 1.50–12.72, *p*-value = 0.01). This study also found that mothers who had an income of less than R 2000 were twice more likely to report PND symptoms (OR = 2.3, CI: 1.17; 4.35, *p*-value = 0.015) than those earning more than R 2000. The source of income was also independently associated with PND symptoms, with mothers who were dependent on others for financial support being almost three times more likely to report PND symptoms (OR = 2.8, CI: 1.15–6.88, *p* = 0.023) than those who were financially independent.

Selected factors that were significantly associated with PND from bivariate analysis were included into a model of multivariate analysis, and the results are reflected in Table 6 below.

## 4. Discussion

The study aimed at identifying social and demographic predictors of PND among HIV-positive women attending Primary Health Care services in a rural district in KwaZulu Natal. The study population was characterized by young women, high unemployment rate, low income of less than R 2000 per month (estimated at 125 USD), and dependent on the government for health and social services. The high proportions of unmarried and unemployed participants were similar to the results of a similar study that was conducted in a rural community in the Western Cape [42]. Being unemployed, most participants were dependent on child support grants, as well as partner and/or family support as the main source of income. The results indicated high prevalence of PND symptoms (42.5%), as measured by the EPNDS with a cut-off point of ≥12. This prevalence falls within a range of previously reported rates within South Africa, which were between 42.2% and 49.3% [6,21,24]. However, it is lower than that reported by a similar study conducted among a population of HIV-positive women in rural settings in Mpumalanga Province, where a prevalence of 48.7% was reported [24]. The difference may be explained by only including HIV-positive pregnant women who had male partners in the previous study, whereas the current study included those who had and those who did not have male partners. Another difference may be explained by differences in community social support structures between the two settings.

From bivariate logistic regression, threats from partner, partner support, partner and family support, being in a relationship, and partner HIV status were statistically significantly associated with PND symptoms. Mothers who did not report being threatened by their partners were four times more likely to report PND symptoms compared with those who were once threatened by their partners. This can be explained by country-level structural stigma, in which women who experience intimate partner violence are so stigmatized that they deny its existence or do not report it [43] However, this phenomenon requires further enquiry. Similar to other studies which found a significant association between intimate partner violence (IPV) and PND [30,43], this study also found that IPV was associated with PND symptoms. Healthcare workers should be trained in screening, detecting, and managing IPV appropriately as they attend to women in postnatal clinics.

A greater proportion of women who are financially dependent on their partners and family were twice times more likely to report PND symptoms than those who were not so dependent. This finding suggests that financial support is only a limited component of social support and therefore does not provide the whole spectrum of social integration and support which is a determinant of mental health [44]. This highlights the importance of social integration for people with HIV, as it influences the overall mental well-being.

The practice of multiple sexual partners is of concern for the HIV-positive population, as such risk behaviors are drivers of HIV infection and reinfection, especially in this province with the highest HIV prevalence in South Africa. The finding of a significant association between adverse life events and PND symptoms was also reported by previous studies [22,45,46]. Adverse life events in the questionnaire included experiencing the death of a loved one, lost employment, moving homes, and separation from a partner or divorce. The adverse life event frequently reported was severe financial crisis.

Both the source of income and household monthly income were found to be independent predictors of PND symptoms among this sample of HIV-positive women, which is similar to other studies [47,48]. However, household monthly income should be interpreted with caution because almost half of the participants did not disclose information regarding this variable. Both source of income and household income highlight low socioeconomic status as a risk factor of postnatal depression [7,12,49,50]. Other than just a financial matter, financial dependence contributes to poor mental outcomes because of associated low self-esteem experienced by the affected individuals.

## 5. Conclusions

The study confirmed the high prevalence of PND among HIV-positive women, with several social and demographic factors associated with PND, which is underdiagnosed and undertreated. Measures to detect depression past the immediate postnatal period among this group of women will be valuable because if left undetected and untreated, depression has been associated with rapid progression of HIV disease, to the detriment of both the mothers and their young children.

### Limitations

The study did not assess previous history of depression [51] and the extent to which the participants utilized antenatal health services, which may provide additional relationships between PND and maternal attributes.

## 6. Recommendations

It is recommended that screening for PND be routinely conducted among all mothers who receive postnatal care in primary healthcare facilities. It is further recommended that future studies include maternal health attributes, both physical and mental, as well as the extent to which the participants attended antenatal care services, as such services may influence PND outcomes.

## Figures and Tables

**Figure 1 healthcare-09-00065-f001:**
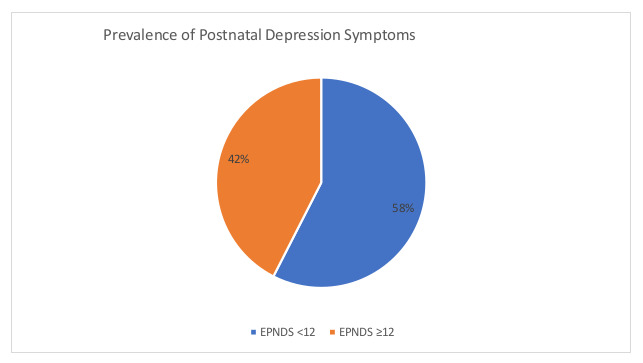
Prevalence of postnatal depression (PND) symptoms.

**Table 1 healthcare-09-00065-t001:** Demographic characteristics of the sample (*n* = 386).

Characteristics	Responses	Value *n* (%)
Maternal age	<25 years	119 (30.9)
	26–35 years	219 (56.7)
	>35 years	48 (12.4)
Marital status	Married	56 (14.5)
	Single/never married	330 (85.5)
Place of residence	Rural	316 (81.9)
	Urban	70 (18.1)
Religion	Christianity	278 (72)
	African traditional	98 (25.4)
	Other	10 (2.6)
Education level	Matric	276 (71.5)
	No matric	110(28.5)
Employment	Not employed	304 (78.8)
	Employed	82 (21.2)
Household monthly income	<R 2000	122 (66)
	R 2001–5000	40 (21.6)
	>R 5000	23 (12.4)
	Did not indicate	201 (52.1)
Source of income	Child support grant	183 (47.4)
	Partner and family support	162 (42)
	Missing	41 (10.6)

**Table 2 healthcare-09-00065-t002:** Social characteristics of the sample.

Variables	Responses	*N* (%)
In a relationship	Yes	338 (87.6)
	No	48 (12.4)
Living with	Alone	29 (7.5)
	Parents/other	257 (66.6)
	Partner	100 (25.9)
Partner financial support	Yes	338 (87.6)
	No	43 (11.1)
	Missing	5 (1.3)
Have other support	Yes	348 (90.2)
	No	38 (9.8)
Received most support from	Partner	74 (19.2)
	Family and friends	251 (65)
	Other	61 (15.8)
Ever threatened by partner	No	328 (85)
	Yes	39 (10.1)
	Missing	19 (4.9)
Male partner has other sexual partners	Yes	109 (28.2)
	No	227 (58.8)
	Do not know	28 (8)
	Missing	22 (5)
Partner drinks alcohol	Yes	192 (49.8)
	No	173 (44.8)
	Missing	21 (5.4)
At least one adverse life event	Yes	139 (36)
	No	244 (63.2)
	Missing	3 (0.08)

**Table 3 healthcare-09-00065-t003:** HIV-related information.

Variable	Responses	*N* (%)
Period since HIV diagnosis	One year	135 (35)
	≥2 years	249 (64.5)
	Missing	2 (0.5)
Partner HIV+	Yes	188 (48.7)
	No	60 (15.5)
	Do not know	125 (32.4)
	Missing	13 (3.4)
Disclosed HIV status to family member	Yes	308 (79.8)
	No	76 (19.7)
	Missing	2 (0.5)
HIV diagnosed during latest pregnancy	Yes	223 (57.8)
	No	158 (40.9)
	Missing	5 (1.3)
Difficulty taking Antiretroviral Therapy (ART) during pregnancy	Yes	54 (14)
	No	301 (78)
	Missing	31 (8%)
Know baby’s HIV status	Yes	217 (56.2)
	No/awaiting results	167 (43.3)
	Missing	2 (0.5)

**Table 4 healthcare-09-00065-t004:** Univariate analysis of sociodemographic factors and PND symptoms.

Variables	Response	Not Depressed *n* = 222 (%)		Depressed *n* = 164 (%)	*p*-Value
Age					
	<25 years	67 (30.1)		52 (31.7)	0.98
	26–35 years	129 (58.1)		90 (54.9)	
	>35 years	26 (11.7)		22 (13.4)	
Marital status					
	Married	35 (15.8)		21 (12.8)	0.42
	Not married	187 (84.2)		143 (87.2)	
Place of residence					
	Rural	187 (84.2)	129 (78.7)		0.16
	Urban	35 (15.8)	35 (21.3)		
Religion					
	Christianity	152 (68.5)	126 (76.8)		0.1
	Traditional	65 (29.3)	33 (20.1)		
	Other	5 (2.6)	5 (3.1)		
Education level					
	Completed matric	172 (77.5)	104 (63.4)		0.01
	Did not complete matric	50 (22.5)	60 (36.6)		
Employment					
	Unemployed	174 (78.4)	130 (79.3)		0.83
	Employed	48 (21.6)	34 (20.7)		
Monthly income					
	<R 2000	100 (73)	22 (45.8)		0.01
	R 2001–5000	26 (19)	14 (29.1)		
	>R 5000	11 (8)	12 (25)		
Source of income					
	Child support grant	114 (57.6)	69 (47)		0.05
	Partner and family support	84 (42.4)	78 (53)		
HIV-related information					
Period since HIV diagnosis					
	1 year	75 (33.9)	60 (36.8)		0.6
	≥2 years	146 (66.1)	103 (63.2)		
Partner HIV+					
	Yes	106 (48.4)	82 (53.3)		0.15
	No	42 (19.2)	18 (11.7)		
Disclosed to family member					
	Yes	179 (81)	129 (79.1)		0.6
	No	42 (19)	34 (20.9)		
HIV diagnosis last pregnancy					
	Yes	133 (60.7)	90 (55.6)		0.31
	No	86 (39.3)	72 (44.4)		
Difficulty taking ART during pregnancy					
Yes		28 (13.3)	26 (17.9)		0.24
No		182 (86.7)	119 (82.1)		
Know baby HIV status					
	Yes	114 (52.5)	103 (47.5)		0.05
	No/awaiting results	108 (64.7)	59 (35.3)		
Partner and social support information					
In a relationship					
	Yes	202 (91)	136 (83)		
	No	20 (9)	28 (17)		0.01
Living with					
	Alone	20 (9)	9 (5.5)		0.30
	Parents/other	142 (64)	115 (70.1)		
	Partner	60 (27)	40 (40.4)		
Partner financial support					
	Yes	203 (92.3)	135 (83.9)		
	No	17 (7.7)	26 (16.1)		0.01
Other social support					
	Yes	207 (93.2)	141 (86)		
	No	15 (6.8)	23 (14)		0.01
Source of most support					
	Partner	48 (22.9)	26 (17.3)		0.2
	Family	135(64.3)	95 (63.3)		
	Friends and other	27 (12.8)	29 (19.3)		
Partner threats					
	Never	205 (95)	123 (81.5)		0.01
	Ever	11 (5)	28 (18.5)		
Partner has other sex partners					
	Yes	160 (74)	67 (45)		
	No	43 (20)	66 (44.3)		0.01
	Do not know	13 (6)	16 (10.7)		
Partner drinks alcohol					
	Yes	115 (53.7)	77 (51)		0.6
	No	99 (46.3)	74 (49)		
Experienced adverse life events					
	Yes	48 (21.6)	91 (56.5)		0.01
	No	174 (78.4)	70 (43.5)		

**Table 5 healthcare-09-00065-t005:** Bivariate analysis of factors associated with PND.

Variables	*p*-Value	Odds Ratio	95% CI
Sociodemographic factors			
Age	0.8	0.9	0.22–3.75
Marital status	0.2	1.6	0.79–3.16
Place of residence	0.16	1.4	0.83–2.52
Religion	0.07	0.7	0.40–1.06
Education level	0.01	0.5	0.31–0.81
Employment	0.83	0.9	0.56–1.59
Monthly income *	0.05	2.8	0.91–11.68
Income source *	0.05	1.5	0.98–2.41
HIV-Related Factors			
Period since HIV diagnosis	0.56	0.9	0.59–1.35
Partner’s HIV status	0.15	1.2	0.96–1.86
Family knows HIV status	0.7	1.1	0.65–1.92
HIV diagnosis when pregnant	0.31	0.8	0.52–1.25
Experienced difficulty taking ART	0.24	1.4	0.75–2.64
Social Support Factors			
Living with whom	0.14	1.7	0.71–4.37
Partner support *	0.01	2.9	1.14–4.69
Other support *	0.01	2.3	1.08–4.80
Most support	0.2	1.4	0.80–2.51
Have partner now *	0.02	2.1	1.08–4.06
Other sexual partners *	0.01	3.5	2.19–5.59
Ever threatened by partner *	0.01	4.2	1.96–9.76
Partner drinks	0.61	0.9	0.58–1.39
Experienced adverse life events *	0.01	4.7	2.94–7.55

* Were statistically significant (*p*-value = 0.01).

**Table 6 healthcare-09-00065-t006:** Multivariate analysis of social and demographic factors associated with PND.

PND Symptoms	Odds Ratio	*p*-Value	95% Conf. Interval
Partner has other sexual partners	4.81	0.007	1.55–14.97
Adverse life event	4.36	0.007	1.50–12.72
HIV diagnosis (pregnant)	0.26	0.013	0.09–0.75
Level of education	0.90	0.582	0.63–1.30
Income	2.25	0.015	1.17–4.35
Source of income	2.81	0.023	1.15–6.88
Know baby HIV status	0.47	0.065	0.21–1.05

## Data Availability

The data presented in this study are available on request from the Sefako Makgatho Health Sciences University. The data are not publicly available because the University does not yet have a platform to avail its data to the public.

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
