# Peer review of "Social and Demographic Factors Associated with Postnatal Depression Symptoms among HIV-Positive Women in Primary Healthcare Facilities, South Africa"

_healthcare, 2021, doi:10.3390/healthcare9010065_

Round 1

Reviewer 1 Report

The "problem" is clearly identified and the research design is appropriate. The limitations of the research are high given the specific context of the research--For HIV mothers attending a primary health care unit.  the use of acronyms is not useful.  for example, line 88 refers to ART.  What does it mean?  Then on line 144 there is a reference to "metric"?  And line 254 it talks about IPV, which means?  I liked the "disclaimers" noted by the authors and the identification of unexpected results, e.g., people who were not "threatened" by domestic partners were more likely to have PND than those that were.  A bit more explanation on that result would be useful.  In that respect, is there any time issues that might have impacted this last result?  If possible, it would be nice to see another sample of non-HIV mothers comparing the independent variables and the rate of PND.  But perhaps that is another story to be told.

Reviewer 2 Report

I congratulate the authors on this study, it is well written. I have a number of minor comments:

1) Revise Figure 1, there is no reason to present this as a 3D image

2) p-values should NEVER be reported as = 0.0!

3) Present 95% CI with ALL summary statistics (eg ORs)

Reviewer 3 Report

This article from Kebogile Elizabeth Mokwena and Nontokozo Lilian Mbatha  describe interesting data associated with postnatal depression, among HIV positive women. 

This manuscript suffers from many limitations that have to be addressed before possible publication.

  • Paragraph 2.1: More detail have to be given on the period of inclusion. 
  • Paragraph 2.4: A cut-off value of 12 has been chosen. The authors give details (has to be included into the discussion part of the manuscript; line 120-125) for the 10 and 11 thresholds without justification for the choice of 12. Please complete.
  • Paragraph 2.4 and 2.6 could be fuse as a unique paragraph as the data are partly redundant.
  • Paragraph 2.7. Have multiple testing correction been considered
  • Paragraph 3.1.:
    • Simplify the understanding of the purpose by indicating Interquartile range.
    • Give more detail about eh equivalence of R 2000, give the median salary for example or the percentile corresponding to this threshold.
    • Have the authors investigated the reason why patients did not indicate the household income?
    • Could the authors compare their observation in their cohort to the global south African population (table 1)?
    • Authors have to explicit what they consider as "partner drinks alcohol"? Abuse or normal consumption? And Could the authors list what they consider as an "adverse life event"? Have these details explained to the patients included in the study?
  • Table 4:
    • P-value could not be equal to 0.00.
    • Place of residence categories for the column "Not depressed" exceed 100%.
    • How many digits have the authors considered? Please standardize (0.047 versus 0.05 for example). It would be better to indicate OR IC95 than p-value.
    • The category "knows baby HIV status" would be of more interest if Yes is subdivided in "Yes HIV pos" and "Yes HIV neg."
  • Paragraph 3.4, if p-value is 0.05 CI95, could not include one for OR.
  • Paragraph 3.5. The author have to consider excluding factors that were not significantly associated with PND in bivariate analysis from multivariate analysis. (example Montly income, income source and partner HIV status, CI95 include 1 so p-value >0.05)

  • Global:
    • Double-space have been placed and have to be modified.
    • Please indicate the county more than geographic region (ie South Africa in place of Kwazulu Natal for example)

Round 2

Reviewer 3 Report

This manuscript is now suitable for publication.